# Jealousy, Violence, and Sexual Ambivalence in Adolescent Students According to Emotional Dependency in the Couple Relationship

**DOI:** 10.3390/children8110993

**Published:** 2021-11-02

**Authors:** Félix Arbinaga, María Isabel Mendoza-Sierra, Belén María Caraballo-Aguilar, Irene Buiza-Calzadilla, Lidia Torres-Rosado, Miriam Bernal-López, Julia García-Martínez, Eduardo José Fernández-Ozcorta

**Affiliations:** 1Department of Clinical and Experimental Psychology, University of Huelva, 21071 Huelva, Spain; belen25ca@gmail.com (B.M.C.-A.); irenebuiza18@gmail.com (I.B.-C.); lidia.torres@dpces.uhu.es (L.T.-R.); miriam.bernal@dpces.uhu.es (M.B.-L.); 2Department of Social, Evolutionary and Educational Psychology, University of Huelva, 21071 Huelva, Spain; 3CEU San Pablo Seville School, 41930 Bormujos, Spain; jgarcia@colegioceusevilla.es; 4Department of Physical Activity and Sports, Center for University Studies Cardenal Spínola CEU, University of Seville Attached Centre-Spain, 41930 Bormujos, Spain

**Keywords:** sexism, gender, violence, emotional dependency, youth, abuse

## Abstract

Background: Emotional dependency in couples involves excessive and dysfunctional emotional bonding. Aims: This work aimed to determine the relationship between violence, jealousy, and ambivalent sexism according to emotional dependence in adolescent student couples. Methods: A cross-sectional study. A total of 234 Spanish adolescents (69.7% female, *M_age_* = 16.77, *SD* = 1.11) participated in the study. Participants completed an ad hoc interview and several validated tests (Partner’s Emotional Dependency Scale, the Ambivalent Sexism Inventory, the Jealousy subscale of the Love Addiction Scale, the Conflict in Adolescent Dating Relationship Inventory). Results: Of the sample, 40.6% indicated high emotional dependence and 14.5% extreme emotional dependence. Differences were observed according to gender (*t* = 3.92, *p* < 0.001), with adolescent boys scoring higher than adolescent girls. Extremely emotionally dependent participants showed differences in both violence (sexual, relational, verbal, and physical) and ambivalent sexism (hostile, benevolent) and jealousy scores. Generating a predictive model of emotional dependence, with the variable jealousy and ambivalent sexism as predictor variables, it was found that jealousy has the greatest predictive and major explanatory capacity (*R*^2^ = 0.297); with an *R*^2^ = 0.334. However, the contribution of the ASI-Hostile subscale was not significant when the ASI-Benevolent subscale was introduced into the model. Further, in a second model where the scores on jealousy and the couple conflict inventory’s subscales were considered as predictors, are again jealousy makes the greatest predictive contribution and shows the greatest explanatory capacity (*R*^2^ = 0.296). It was found that the contribution is significant only for the predictive capacity of Sexual Violence and Relational Violence. In this sense, the educational context is one of the propitious places to detect and correct behaviors that may be indicative of potentially unbalanced and unbalancing relationships for adolescents.

## 1. Introduction

Emotional dependence on a partner implies an excessive and dysfunctional emotional attachment to the other person [1]. A person is considered dependent when they perceive the current and past balance of their stable partner relationship to be negative and considers ending the relationship but feels unable to do so even in the absence of economic dependence or threats to remain in the relationship [2]. The relationships between emotional dependence and the gender of the partners are unclear. While some studies have shown that men score higher on emotional dependence [3]; others have found no differences between men and women [4,5].

Emotional dependence may result in a series of negative emotional consequences, including anxious-depressive symptoms, obsessive thoughts, sleep disturbances, and withdrawal from social relationships and leisure activities [6,7]. This dependence is also characterized by a skewed perception of reality, an intolerance of loneliness, and an inner emptiness [1]. The relationship, regardless of its quality, takes priority over any other activity in the life of the person concerned [8]. Such is its importance, adolescents in unhealthy relationships are not only at greater risk of substance use but are also more likely to engage in behaviors that could put them at risk of contracting sexually transmitted diseases [9,10].

Emotional dependence can predispose an individual towards showing aggressive behavior towards a partner [11], which is mainly perpetrated by males [5,12]. Emotional dependence has been considered one of the main precursors of the main precursor of intimate partner violence [13] and is thought to be at the root of violent dating relationships. Such dependence may also increase tolerance to the abuse received and this could make it more difficult to end the relationship [11,14].

By the time young people enter high school, 72% of adolescents have been involved in a dating relationship [15], and between 9 and 38% of these adolescents report having experienced dating violence in the past year [9,16]. Moreover, teen dating violence is often associated with negative physical and mental health consequences [9,15,17]. Therefore, preventive measures in school, social, and family settings are essential [17,18,19].

Dating violence has been defined as the attempt to dominate and control (physically, psychologically, or sexually) the other partner, causing damage to the relationship [20]. Intimate partner violence is a heterogeneous, multi-causal, and gender-independent phenomenon, although the most serious consequences are suffered by women [21]. There are several types of intimate partner violence: verbal, relational, sexual, psychological, and physical [20,22,23]. Adolescents with parents who display aggressive intimate partner behaviors are more likely to engage in aggressive behaviors in their romantic relationships [24,25]. Similarly, young people with friends who exhibit aggressive intimate partner behaviors may come to normalize and justify the use of aggression to resolve intimate partner conflicts [26].

Emotional dependence on a partner can lead to jealousy-related effects [27,28]. Jealousy is a social emotion generated by a threat or actual loss of a valued relationship in a triadic context. Jealousy is often generated by the existence of a rival for a partner’s attention [29]. This rival, human or not, can be real or imagined [30]. Jealousy is considered an unpleasant emotion and has been linked to negative relationship outcomes, including relationship conflict, domestic violence, and divorce [31,32,33]. The relationship between emotional dependence and jealousy in individuals could be the result of a threat to the relationship that may appear more costly to these individuals [27,28]. Regardless of the source of dependence in the relationship, it is observed that the intensity of jealous reactions increases with increasing dependence [29]; while emotional dependence is an important predictor of jealousy, it is far from the only predictor [32].

Emotional dependence and jealousy are related to ambivalent sexism [34]. Ambivalent sexism refers to the coexistence of negative and positive affect and attitudes towards women [35]. This construct encompasses two types of attitudes; hostile sexism, which refers to the belief that women are inferior to men, and benevolent sexism, which is more covert and subtle, and expresses men’s desire to protect women while emphasizing their ability to perform tasks that conform to roles associated with the traditional female stereotype.

From Ambivalent Sexism Theory (AST), three dimensions have been distinguished within both hostile sexism and benevolent sexism [36]. The first is paternalism and relates to the distribution of power: in hostile sexism, paternalism is of a dominating type, whereas for benevolent sexism, it is of a protective type. The second dimension concerns gender differentiation, either competitive (hostile sexism) or complementary (benevolent sexism). Finally, the third dimension of sexism concerns sexuality. Women either lack sexuality or have powerful sexuality that makes them dangerous to men (hostile sexism).

On the other hand, heterosexual partnerships are essential to achieve true happiness (benevolent sexism). From the framework of AST, a speculative model of the development of gender bias is proposed to explain how this evolves from an aggressive form in childhood to a set of ambivalent attitudes in adulthood towards people of the other sex [34]. The critical period for change is puberty since it is at this point that, along with gender differentiation and power, heterosexual romantic impulses come into play [34,36].

In the adolescent population, it was observed that benevolent sexism decreased with increasing age [37,38]. Similarly, in males, the older the age, the lower the level of hostile sexism. This result, a priori somewhat contradictory to the predictions of Glick and Hilt [34] on the development of sexism, can be explained by considering that the older the age the greater the awareness of the injustice of sexism. On the other hand, Glick et al. [39] found that education was associated with a lower sexist attitude (benevolent and hostile). At the same time, it has been observed that when gender stereotypes are violated, women are perceived as competent but unfriendly [40]. Therefore, it seems that while low-status female subtypes (e.g., housewives) are dehumanized, high-status female subtypes are exposed to reactions for violating the prescriptions of feminine kindness [41].

Sexist attitudes (Hostile Sexism and Benevolent Sexism) show important relationships with aggression against women within intimate partner relationships [42]. Violence against women within couples is a widespread phenomenon, with psychological violence being the most prevalent [43] involving insults, humiliation, and controlling behaviors that produce psychological damage [44]. Psychological violence is often thought to precede, and thus to be an important risk factor for, physical violence [43]. However, psychological violence often occurs independently of other violence such as physical and sexual violence and can often arise during routine relationship interactions [45].

In this context, the present study aims to expand existing knowledge on emotional dependence within adolescent dating relationships and its interaction with violence or abuse, jealousy, discriminatory attitudes, and behavior towards partners. As a first working hypothesis, we expect to find an absence of differences between adolescent boys and girls in the scores on emotional dependence on the partner. On the other hand, as a second hypothesis, it is expected to observe that people with higher scores on emotional dependence will show greater violence in all the dimensions analyzed. As a third hypothesis, it is expected to find that people with higher scores on emotional dependence will reflect greater sexual ambivalence and, as a fourth hypothesis, it is predicted that people with higher scores on emotional dependence will report higher scores on the evaluation of jealousy.

## 2. Materials and Methods

### 2.1. Study Design, and Participants

A cross-sectional study. The sample comprised 234 Spanish adolescents (69.7%, female, *M_age_* = 16.77, *SD* = 1.11; *Min* = 15 and *Max* = 19); who were high-school students. Thirty-eight percent (38%) acknowledged that they currently have a partner and 38.50% acknowledged being jealous (see Table 1).

Using the G* Power 3.1 (2017) sample size calculator, for an *alpha* error of 0.01, a *power* of 0.99, and an effect size of 0.1 for Linear multiple regression tests (Fixed model, *R*^2^ increase) with six predictors, a group size of 244 was estimated to be necessary. Along these lines, the number of people surveyed was increased to 250 people. However, 16 people did not meet the criteria for inclusion in the study.

### 2.2. Study Variables and Instruments

The gender and year of birth of each participant were recorded. They were then asked to answer a series of questions about their relationships, starting with: Do you have a boyfriend/girlfriend? (Yes/No); if they answered “Yes” to the question, they had to indicate how long they had been together (Yes/No); what has been your longest relationship over time; how many partners have you had; and do you consider yourself jealous? (Yes/No).

To determine the degree of emotional dependency, they were asked to complete the Partner’s Emotional Dependency Scale (PEDS) [4]. This scale is unidimensional and consists of 22 items with a Likert-type response format (0—Totally False to 4—Totally True). In addition, four cut-off points are presented according to the degree of dependence (<9 points: “Low Dependence”; 10 to 21 points: “Moderate Dependence”; 22 to 36 points: “High Dependence”; >37 points: “Extreme Dependence”. The Cronbach’s alpha obtained in this work was 0.860.

To establish discriminatory attitudes and behaviors towards women, the participants completed the Ambivalent Sexism Inventory (ASI) [46], an instrument designed, developed, and validated for the Spanish adolescent population based on the Ambivalent Sexism Inventory [35]. It is organized into ten items for the “Hostile Sexism” subscale and another ten items for the “Benevolent Sexism” subscale. The response range is between 1—“Strongly Disagree” and 6—“Strongly Agree.” In this study, Cronbach’s alpha coefficient was 0.868 (Hostile Sexism = 0.828 and Benevolent Sexism = 0.795).

The students completed the Jealousy subscale of the Love Addiction Scale [47]. This subscale is composed of four items, with a Likert-type response format (1—“Strongly disagree” to 5—“Strongly agree”) and in this study, the subscale achieved a Cronbach’s alpha of 0.841.

The Conflict in Adolescent Dating Relationships Inventory (CADRI) [23], adapted to the Spanish population [48] in its revised version [49], was used to evaluate violence within adolescent couples. The inventory consists of 70 items divided into two subscales (violence perpetrated and received). In turn, each of these subscales is subdivided into five factors: physical violence, relational violence, sexual violence, verbal violence, and conflict resolution. The response modality of this inventory is a scale with four response options (0—“Never” to 3—“Frequently”); obtaining three scores for each factor (one score for perpetrated, one score for received, and one total score). For the factor “Verbal Violence,” the Cronbach’s alpha obtained for the total factor was 0.874; for Perpetrated, this was 0.748, and for Received, this was 0.809. For the Conflict Resolution factor, the Cronbach’s alpha was 0.844, for Perpetrated it was 0.717, and for Received was 0.725. For the Sexual Violence factor, Cronbach’s alpha for the total factor was 0.721, for Perpetrated it was 0.460 and for Received it was 0.619. For the Relational Violence factor, Cronbach’s alpha for the total factor was 0.787, for Perpetrated this was 0.639, and for Received this was 0.740. Finally, for the Physical Violence factor, Cronbach’s alpha for the total factor was 0.816, for Perpetrated this was 0.606, and for Received this was 0.743.

### 2.3. Procedure

The data were collected in various school classrooms in joint application sessions lasting around 45 min. The research was presented in all classes where the school management reported that there were students over 15 years of age. The schools were selected by non-probabilistic sampling according to accessibility and availability criteria. For underage participants, the school administration provided explicit consent from parents or legal guardians; and in the case of over-age students, they filled in the informed consent form. At the beginning of the sessions, students were informed of the voluntary nature of their participation, the guarantee of their anonymity at all times, and that their responses would be used exclusively for research purposes. The students were informed that this was an investigation to know the characteristics that young adolescents show in their relationships; there is no good/bad or true/false information, it is only about knowing how you behave when you are in a relationship.

Inclusion criteria were: being older than 15 years and accepting to participate; signing the informed consent form; having a current relationship of at least one month’s duration or having been in one recently, with no more than two months since the breakup; completing all the instruments; not having a disability that prevents them from understanding and answering the various questionnaires; not having parental limitations to participation. Approval for this study was obtained from the University Bioethics Committee (5 July 2020).

### 2.4. Statistical Methods

Statistical analyses were carried out using the SPSS statistical package (IBM version 20.0, SPSS Inc., Chicago, IL, USA). Descriptive statistics (means, standard deviation, frequencies, and percentages) were used to describe the different variables. Student’s *t*-tests for independent samples were used to compare categorical and quantitative variables, and effect sizes were estimated using Cohen’s d (<0.2.—Small Effect Size; around 0.5.—Medium Effect Size and >0.8.—Large Effect Size). For comparisons of categorical variables, *Chi*^2^ (*x*^2^_(*df,n*)_) and its corresponding effect sizes (Phi and Cramer’s V) were used. For simple *ANOVA* tests, partial eta squared estimates (*ŋ*^2^*_p_*) were used to calculate effect sizes (0.01 ≤ *ŋ*^2^*_p_* < 0.06 = a small effect size, 0.06 ≤ *ŋ*^2^*_p_* < 0.14 = a medium effect size, and *ŋ*^2^*_p_* ≥ 0.14 = a large effect size) [50,51].

Finally, stepwise regression analyses were conducted to assess the predictive ability of scores for the Jealousy variables, and the ASI and CADRI subscales for the Emotional Dependence in Partners test score.

## 3. Results

The mean age of the sample was 16.77 years (*SD* = 1.11, *Min* = 15, *Max* = 19); with no significant difference between women (*M_age_* = 16.8, *SD* = 1.11) and men (*M_age_* = 16.72, *SD* = 1.12) (*t* = 0.501, *p* = 0.617). 18.8% of the participants said they were 15 years old, 17.5% were 16 years old, 32.5% said they were 17 years old, 29.9% said they were 18 years old, and 1.3% said they were 19 years old. Thirty-eight percent of the sample acknowledged that they currently have a partner, with no difference between men and women (see Table 1).

Table 2 shows that men and women score equally on the jealousy scale. However, men score higher than women on the two subscales of ambivalent sexism, with a large effect size for hostile sexism and the total test score and a medium effect size for benevolent sexism. There were also significant differences between men and women in the scores obtained on the scale of emotional dependence on the partner. Women tend to score lower on the emotional dependence category while men tend to be in the extreme emotional dependence category (Cramer’s V = 0.233).

When comparing the scores obtained between males and females on the CADRI (Table 3), differences were only found for sexual violence. Men obtained higher scores (with small effect sizes) on conflict resolution received, where women obtain a higher score with a medium effect size.

Looking at the participants grouped into the four categories of the emotional dependence on the partner scale (Table 4), the groups show differences in all dimensions of the Conflict in Adolescent Dating Relationship Inventory, except for the physical violence subscale and the three conflict resolution scores.

After carrying out multiple comparisons, using the Bonferroni post hoc test to compare the categories in emotional dependence and the scores obtained on the various subscales of the CADRI, it can be observed that for sexual violence, differences were found in the Perpetrated, Received, and total dimensions. In Perpetrated, differences were observed between a < d (*p* = 0.009) and between c < d (*p* = 0.016). In sexual violence received, the differences are found between a < d (*p* = 0.004) and between b < d (*p* = 0.021). For the total score on sexual violence, differences were found between a < d (*p* = 0.001) and between c < d (*p* = 0.035). Regarding relational violence, differences were found in violence between a < d (*p* = 0.027), b < d (*p* = 0.003) and between c < d in a residual way (*p* = 0.059). In the case of relational violence received, differences were observed between a < d (*p* < 0.001), b < d (*p* < 0.001) and c < d (*p* < 0.001). For the total relational violence score, the differences are between a < d (*p* < 0.001), b < d (*p* < 0.001) and between c < d (*p* < 0.001). Finally, differences can be observed in Received and total physical violence. For Received, there are differences between a < d (*p* = 0.011) and between b < d (*p* = 0.004). For the total score on the physical violence subscale, the differences are between a < d (*p* = 0.024) and between b < d (*p* = 0.058). With respect to the subscale of verbal violence, significant differences are between a < c (*p* = 0.05), a < d (*p* = 0.008), b < c (*p* = 0.036) and b < d (*p* = 0.006). For verbal violence received, the significant differences are a < c (*p* = 0.012), a < d (*p* < 0.001), b < c (*p* = 0.003), b < d (*p* < 0.001) and c < d (*p* < 0.001). In relation to the total score on verbal violence, differences were observed between groups a < c (*p* = 0.013), a < d (*p* < 0.001), b < c (*p* = 0.005), b < d (*p* < 0.001), and c < d (*p* = 0.007). Finally, looking at the conflict resolution subscale, the Bonferroni post hoc test found that the three scores (Perpetrated, Received, and total) were equivalent across the groups.

Table 5 shows the scores obtained on the jealousy and ambivalent sexism scales according to the grouping of the participants based on their emotional dependence. Again, for both jealousy and ambivalent sexism, there were significant differences between the groups, with large and medium effect sizes.

After conducting multiple comparisons using the Bonferroni test, for the jealousy scores, differences were observed for a < c (*p* = 0.001), a < d (*p* < 0.001), b < c (*p* < 0.001), b < d (*p* < 0.001) and c < d (*p* < 0.001). For the ambivalent sexism scores, differences were found in hostile sexism for groups a < d (*p* < 0.001) and b < d (*p* = 0.002). For benevolent sexism, differences were found between groups a < d (*p* = 0.004), b < c (*p* = 0.017) and b < d (*p* < 0.001). Finally, for the total ambivalent sexism score, significant differences were found between a < c (*p* = 0.033), a < d (*p* < 0.001), b < c (*p* = 0.044) and b < d (*p* < 0.001).

After performing a stepwise linear regression analysis including the “jealousy” subscale of the ASI and the CADRI subscales as predictor variables for the total score on Emotional Dependency on the partner, it can be observed that after generating two models (Table 6) the jealousy scores make a greater contribution to the predictive capacity of the models for the emotional dependency of the partner. In Model 1, where jealousy scores and the ASI subscales are considered, jealousy alone has a predictive capacity (*β* = 0.545) that is much greater than the rest of the variables, explaining 33.4% of the variance when the ASI-Hostile and ASI-Benevolent variables are added. However, the contribution of the ASI-Hostile subscale is not significant (*p* = 0.267) when the ASI-Benevolent subscale is added.

In Model 2, where the jealousy scores and the CADRI subscales are considered, it is observed that it is the jealousy variable which again makes the greatest predictive contribution (*β* = 0.544) and explains most of the variance (*R*^2^ = 0.296). When the various subscales of the CADRI are incorporated into the model, the contribution is significant only for the predictive capacity of Sexual Violence (reaching a maximum of *β* = 0.181) and Relational Violence (*β* = 0.194).

## 4. Discussion

This study aimed to expand our knowledge about emotional dependence on partners in the context of dating relationships in young adolescents and the interactions that may occur with violence or abuse, jealousy, discriminatory attitudes, and behaviors towards partners. As a first working hypothesis, we expected to find an absence of differences between adolescent boys and girls in scores on emotional dependence on a partner. Our second hypothesis predicted that those with higher scores on emotional dependence would show greater violence in all its dimensions. As a third hypothesis, we expected to find that people with higher scores on emotional dependence would show greater sexual ambivalence, and, as a fourth hypothesis, it was predicted that people with higher scores on emotional dependence would report higher scores on jealousy. Finally, we sought to develop predictive models of emotional dependence based on jealousy, violence, and sexual ambivalence.

The first hypothesis was not confirmed by the results obtained. The data have shown that it is boys, as opposed to girls, who show higher scores on emotional dependence on their partner. While girls tend to show low emotional dependence, boys tend to be in the extreme emotional dependence categories. These results are consistent with those obtained in a study on 1092 young adolescent students. Emotional dependence, early maladaptive schemas, and variables associated with partner relationships were analyzed, finding that emotional dependence was significantly higher in males [3].

Similarly, our findings are consistent with those studies that have controlled for the age variable and have shown that male participants aged 17 to 19 years show higher emotional dependency scores [52]. However, our results contradict those studies that report no gender differences in the degree of emotional dependence on partners [4,5]; both in clinical and general populations [53]. The literature also shows that women obtain higher scores on scales that have generally been understood as measuring demanding behaviors or that denote emotional dependence (effective expression of the partner and fear of loneliness), considering that these behavioral expressions of emotional dependence may be more visible in women than in men, who in turn reported higher scores in attention-seeking [54,55,56]. In a similar vein, studies have been reviewed where significant differences can be observed when comparing men’s and women’s scores on subscales of destructive overdependence and healthy dependence, with higher scores in women [57].

Our second hypothesis—where we expected to observe that people with higher scores on emotional dependence would show greater violence in all its dimensions—has been supported by the data from our study, although not with all the subscales of the CADRI. In the Conflict in Adolescent Dating Relationship Inventory, we found an absence of differences in the conflict resolution scale (Perpetrated, Received, and total) and in the physical violence subscale. Young people classified as extremely emotionally dependent scored highest on all three measures of sexual violence, relational violence, verbal violence, and the subscale of received and total physical violence. These differences were observed between the emotionally dependent groups (low, moderate, high, and extremely), with the low emotional dependence group scoring the lowest on the various scales of the test. It should be noted that no gender differences were observed; contrary to what has been indicated in the literature. Given the gender differences in social roles, males express their dependence in a more subtle way, so that for the male in the relationship, there co-exists both an emotional need and intense aggressiveness, that is, the emotionally dependent male not only needs but also belittles his partner [58,59].

Young people classified as extremely emotionally dependent show a high score in both relational violence received and verbal violence received; while they also show differences concerning physical violence received but not perpetrated. These results support those of other studies in the literature. For instance, it has been reported that victimization is more intense in those who present greater emotional dependence than those who have not suffered violence in an intimate partner relationship [60]. In this sense, [61] points out that one of the dimensions closely related to intimate partner violence is the behavioral pattern of emotional dependents [5,42]. Emotional dependence has been considered as a precursor of intimate partner violence [13,43], noting that emotional dependence would be at the root of the maintenance of violent dating relationships, increasing tolerance to the abuse received and making it difficult to end the relationship [11,14]. Thus, a positive correlation has been identified between emotional dependence and proactive aggression [45,62], with gender differences where physically violent men showed extremely high levels of emotional dependence compared to non-violent subjects in both happy and unhappy relationships [58]. However, it is also necessary to consider that victims often do not interpret aggression as such but instead trivialize it; in this sense, it is interesting to compare perceptions of subtle and overt violence [43,63]. This trivialization of violence could also explain the lack of differences in physical violence observed in the present study. The aggressor may consider certain violent behaviors as normalized and not acts of aggression [40,41,62].

As predicted by our third hypothesis, it has been found that the scores on sexual ambivalence increase as participants move through the categories of emotional dependence, from low dependence to extreme dependence. Similarly, boys obtained significantly higher scores on hostile sexism, benevolent sexism, and the total score of the ambivalent sexism inventory. These results are consistent with findings that show existing differences in hostile sexism as a function of gender, with higher scores in the male group, which also translates into higher scores in total sexism [34,37,38,52,64]. However, these studies showed these gender differences only in the 17 and 18 age groups, when dependence takes the highest values in the male group [34,52]. Thus, controlling for the age variable makes it plausible that sexism appears as a reaction to dependency needs in adolescent boys at these ages. Furthermore, these results are partially consistent with those of other research since, although hostile sexism is usually higher in adolescent boys [34,64], emotional dependence scores do not differ according to gender [11]. However, studies that control for age have reported higher emotional dependence scores in boys as a function of their age [3], as was the case in our study.

The literature shows that for emotionally dependent people, it is impossible to imagine their existence without their partner [65] and this can establish asymmetrical relationships by engaging in aggressive and sexist behavior. It appears that hostile sexism has a negative emotional charge, in which men adopt an inferior view of women, with the former dominating the situation. Sexist attitudes do not allow women to be seen as equal, even though they have similar responsibilities [34,40,41,66].

Our fourth hypothesis predicted that people with higher scores on emotional dependence would also score high on jealousy. This hypothesis has been supported by our data, since participants classified as having a higher level of emotional dependence on their partner obtained higher scores on jealousy, except for the groups of low and moderate emotional dependence, which are similar.

Previous research has already indicated that emotional dependence on a relationship affects jealousy since it is possible that threats to the relationship are interpreted as more costly for these individuals [27,28]. Regardless of the source of dependence in the relationship, the intensity of jealousy reactions increased with increasing emotional dependence [29]. Although dependence was found to be an important predictor of jealousy, it is far from the only relevant variable. In this regard, it has been suggested that the model of the self, and the model that others have of the self, can have an impact on jealousy beyond dependence [32]. In particular, individuals who were secure and had more positive models of themselves indicated lower levels of jealousy. Having positive attachment models also appeared to be a valuable asset for buffering against inadequacy and dependence and, in turn, reducing feelings of romantic jealousy [32].

Our results indicate no significant differences between boys and girls on the scores of the jealousy subscale or in their opinions when defining themselves as jealous. However, our data support more recent findings that have questioned the consistency, validity, and robustness of these results indicating such differences in jealousy [67,68,69], finding no differences between boys and girls in their jealousy-related behavior.

The present study has several limitations. First, this is a correlational study, and therefore it is not possible to draw any firm conclusions regarding causal relationships between the variables studied. Moreover, our sample was composed of adolescent students, and it would be interesting to consider young people who have dropped out of school. It should also be noted that no information was collected on the family structure and relationships between parents in the home environment, while there was no record of peer group relationships and attitudes towards violence in the context of a couple relationship. Based on the data provided by the literature, it could be interesting to control and analyze the role played by variables such as self-esteem, attributional styles, social networks, and contacts, when considering emotional dependence.

It should also be noted that the questionnaire used to evaluate the violence perpetrated-received (CADRI) does not detect how each participant interprets their behavior towards their partner. And although this questionnaire asks about the discomfort caused by each behavior of their partner, this does not imply that it captures the participant’s perception of violence. In this sense, when using the CADRI, participants had to base their responses on their longest relationship in the last year, referring only to the last twelve months, and the same partner. This could be problematic as although the participants may have been in a relationship, this was not the one that is most relevant for the purposes of the present study.

## 5. Conclusions

Throughout the life cycle, it is necessary to develop healthy, respectful, and emotionally balanced relationships between partners. However, this is especially relevant in adolescence, as an early stage of learning and initiation of emotional interactions within the framework of a couple, since it will allow avoiding, during adulthood, future situations of intergender violence. In addition, it has been observed that emotional dependence is often associated with alterations in the mental health and quality of life of those who live independent contexts. In this sense, the family framework and the educational context are propitious places to detect and correct behaviors that may be indicative of potentially unbalanced and unbalancing relationships for young people.

## Figures and Tables

**Table 1 children-08-00993-t001:** Characteristics of couple relationships according to the gender of the participants.

	Female 163 (69.7)	Male 71 (30.3)	Total 234		*p*
Has a partner				*x*^2^_(1,234)_ = 1.376	0.241
Yes	66 (40.50)	23 (32.40)	89 (38.00)		
No	97 (59.50)	48 (67.60)	145 (62.00)		
Current partner time	5.81 (10.30)	3.70 (7.54)	5.17 (9.58)	*t* = 1.747	0.082
Longer relationship	11.90 (10.57)	10.38 (10.04)	11.44 (10.41)	*t* = 1.023	0.307
How many couples	1.99 (1.26)	2.34 (1.673)	2.09 (1.40)	*t* = 1.581	0.117
Do you consider Jealous				*x*^2^_(1,234)_ = 1.585	0.208
Yes	67 (41.10)	23 (32.40)	90 (38.50)		
No	96 (58.90)	48 (67.60)	144 (61.50)		

Note: Frequency and percentage of cases for categorical variables. Mean and standard deviation for continuous variables; Time with current partner (months): How long have you been with your current partner; Longest relationship (months): What has been your longest relationship; How many partners: How many partners have you had; Do you consider yourself jealous?

**Table 2 children-08-00993-t002:** Scores obtained on the jealousy subscale, emotional dependence, ambivalent sexism, and categories of the partner emotional dependence scale according to the gender of the participants.

	Female 163 (69.70)	Male 71 (30.30)	Total 234	*t*	*p*	*Cohen’s d*
Jealousy	6.18 (3.11)	6.80 (3.25)	6.49 (3.18)	1.39	0.172	0.19
ASI						
Hostile	1.84 (0.39)	2.59 (0.82)	2.07 (0.81)	7.134	<0.001	1.16
Benevolent	2.30 (0.85)	2.80 (0.85)	2.45 (0.88)	4.139	<0.001	0.59
Total	2.07 (0.67)	2.70 (0.73)	2.26 (0.74)	6.394	<0.001	0.90
PEDS	22.35 (11.57)	28.99 (12.68)	24.36 (12.27)	3.92	<0.001	0.55
Categories PEDS				*x*^2^_(3,234)_*=* 12.718	0.005	
Low	19 (11.70)	1 (1.40)	20 (8.50)			
Moderate	64 (39.30)	21 (29.60)	85 (36.60)			
High	62 (38.00)	33 (46.50)	95 (40.60)			
Extreme	18 (11.00)	16 (22.50)	34 (14.50)			

Note: Frequency and percentage of cases for categorical variables. Mean and Standard Deviation for continuous variables; PEDS—Partner’s Emotional Dependency Scale; ASI—Ambivalent Sexism Inventory; Jealousy—Jealousy subscale of the Love Addiction Scale.

**Table 3 children-08-00993-t003:** Subscale scores on the Conflict in Adolescent Dating Relationship Inventory (CADRI) according to the gender of the participants.

	Female 163 (69.70)	Male 71 (30.30)	Total 234	*t*	*p*	*Cohen’s d*
Sexual Violence						
Perpetrated	4.53 (1.04)	4.94 (1.59)	4.75 (1.34)	1.99	0.049	0.31
Received	4.91 (1.70)	5.01 (1.49)	4.96 (1.60)	0.45	0.650	0.06
Total	9.44 (2.49)	9.96 (2.74)	9.7 (2.62)	1.41	0.159	0.20
Relational Violence						
Perpetrated	5.25 (0.71)	5.51 (1.76)	5.38 (1.34)	1.33	0.186	0.19
Received	5.95 (1.99)	6.42 (2.33)	6.19 (2.17)	1.58	0.140	0.22
Total	11.20 (2.44)	11.90 (3.55)	11.57 (3.05)	1.57	0.115	0.23
Physical Violence						
Perpetrated	7.44 (1.06)	7.52 (1.52)	7.48 (1.31)	0.46	0.648	0.06
Received	7.68 (1.78)	7.59 (1.43)	7.64 (1.61)	0.37	0.708	0.06
Total	15.12 (2.56)	15.11 (2.86)	15.12 (2.71)	0.03	0.980	0.00
Verbal Violence						
Perpetrated	14.64 (4.00)	14.31 (4.08)	14.54 (4.02)	0.58	0.560	0.08
Received	15.25 (4.88)	15.10 (4.49)	15.21 (4.76)	0.23	0.822	0.03
Total	29.90 (8.29)	29.41 (8.13)	29.75 (8.23)	0.42	0.678	0.06
Conflict Resolution						
Perpetrated	28.79 (4.64)	28.45 (5.15)	28.68 (4.79)	0.49	0.624	0.07
Received	27.48 (4.76)	25.52 (4.87)	26.89 (4.87)	2.88	0.004	0.41
Total	56.27 (8.94)	53.97 (9.26)	55.55 (9.09)	1.79	0.075	0.25

Note: Frequency and percentage of cases are given for categorical variables. Means and standard deviations are presented for continuous variables.

**Table 4 children-08-00993-t004:** Subscale scores on the CADRI—Conflict in Adolescent Dating Relationship Inventory according to categories of the Partner’s Emotional Dependency Scale (PEDS).

	Low (a)	Moderate (b)	High (c)	Extreme (d)	*F* _(3,233)_	*p*	*ŋ* ^2^ * _p_ *
Sexual Violence							
Perpetrated	4.10 (0.31)	4.46 (0.97)	4.76 (1.26)	5.21 (1.82)	4.68	0.003	0.06
Received	4.40 (1.00)	4.59 (1.18)	5.01 (1.60)	5.94 (2.46)	6.80	<0.001	0.08
Total	8.50 (1.05)	9.05 (2.00)	9.77 (2.66)	11.15 (3.43)	7.30	<0.001	0.09
Relational Violence							
Perpetrated	5.05 (0.22)	5.14 (0.81)	5.35 (0.73)	5.88 (2.04)	4.70	0.003	0.06
Received	5.35 (1.20)	5.67 (1.50)	5.96 (1.71)	7.97 (3.50)	12.82	<0.001	0.14
Total	10.40 (1.40)	10.82 (2.04)	11.31 (2.21)	13.85 (4.83)	11.99	<0.001	0.14
Physical Violence							
Perpetrated	7.30 (0.22)	7.36 (1.25)	7.48 (1.16)	7.76 (1.50)	1.01	0.390	0.01
Received	7.05 (0.22)	7.36 (1.13)	7.74 (1.50)	8.50 (3.02)	4.91	0.003	0.06
Total	14.35 (0.81)	14.73 (2.30)	15.22 (2.33)	16.26 (4.25)	3.45	0.017	0.04
Verbal Violence							
Perpetrated	12.65 (3.62)	13.59 (3.86)	15.20 (3.83)	16.21 (4.26)	6.26	<0.001	0.08
Received	12.45 (3.61)	13.47 (3.49)	15.72 (4.35)	19.74 (5.73)	20.94	<0.001	0.21
Total	25.10 (7.03)	27.06 (7.08)	30.92 (7.65)	35.94 (8.96)	14.29	<0.001	0.16
Conflict Resolution							
Perpetrated	27.80 (5.44)	27.84 (5.18)	29.04 (4.50)	30.32 (3.69)	2.68	0.048	0.03
Received	27.10 (5.33)	26.53 (5.31)	27.34 (4.53)	26.41 (4.45)	0.54	0.655	0.01
Total	54.90 (10.52)	54.36 (10.23)	56.38 (8.40)	56.74 (6.59)	0.97	0.406	0.01

Note: Means and standard deviations are presented.

**Table 5 children-08-00993-t005:** Test scores on jealousy, hostile sexism, and ambivalent sexism according to emotional dependency categories.

	Low (a)	Moderate (b)	High (c)	Extreme (d)	*F* _(3,233)_	*p*	*ŋ* ^2^ * _p_ *
Jealousy	4.15 (0.49)	5.15 (1.92)	6.79 (2.85)	9.56 (4.45)	26.13	<0.001	0.25
ASI							
Hostile	1.64 (0.68)	1.93 (0.68)	2.12 (0.82)	2.52 (0.96)	6.80	<0.001	0.08
Benevolent	2.09 (0.85)	2.20 (0.77)	2.58 (0.87)	2.91 (0.92)	7.92	<0.001	0.09
Total	1.87 (0.69)	2.07 (0.63)	2.35 (0.71)	2.72 (0.86)	9.56	<0.001	0.11

Note: Means and standard deviations are presented; ASI—Ambivalent Sexism Inventory, Jealousy—Jealousy subscale of the Love Addiction Scale.

**Table 6 children-08-00993-t006:** Linear stepwise regression of predictor variables (Love Addiction Scale jealousy subscale, ASI subscales and CADRI subscales) on Partner’s Emotional Dependency.

**MODEL 1**		** *β* **	** *t* **	** *p* **	** *R* ^2^ **	**Δ*R*^2^**	** *p* **	** *F* **	** *p* **
F1	Jealousy	0.545	9.908	<0.001	0.297	0.297	<0.001	*F*_(1,233)_ = 98.178	<0.001
F2	Jealousy	0.51	9.175	<0.001	0.322	0.024	0.004	*F*_(2,233)_ = 54.754	<0.001
	ISA-Hostile	0.16	2.874	0.004					
F3	Jealousy	0.481	8.622	<0.001	0.334	0.021	0.007	*F*_(3,233)_ = 39.923	<0.001
	ISA-Hostile	0.071	1.112	0.267					
	ISA-Benevolent	0.175	2699	0.007					
**MODEL 2**		** *β* **	** *t* **	** *p* **	** *R* ^2^ **	**Δ*R*^2^**	** *p* **	** *F* **	** *p* **
F1	Jealousy	0.544	9.948	<0.001	0.296	0.296	<0.001	*F*_(1,232)_ = 96.982	<0.001
F2	Jealousy	0.492	8.937	<0.001	0.342	0.046	<0.001	*F*_(2,232)_ = 59.769	<0.001
	Sexual Violence	0.221	4.023	<0.001					
F3	Jealousy	0.443	7.865	<0.001	0.367	0.025	0.003	*F*_(3,232)_ = 44.321	<0.001
	Sexual Violence	0.159	2.748	0.006					
	Relational Violence	0.182	3.029	0.003					
F4	Jealousy	0.449	7.9	<0.001	0.369	0.002	0.414	*F*_(4,232)_ = 33.360	<0.001
	Sexual Violence	0.181	2.834	0.005					
	Relational Violence	0.194	3.135	0.002					
	Physical Violence	−0.053	−0.818	0.414					
F5	Jealousy	0.424	7.202	<0.001	0.376	0.007	0.109	*F*_(5,232)_ = 27.390	<0.001
	Sexual Violence	0.17	2.663	0.008					
	Relational Violence	0.159	2.436	0.016					
	Physical Violence	−0.068	−1.038	0.3					
	Verbal Violence	0.106	1.607	0.109					
F6	Jealousy	0.431	7.349	<0.001	0.385	0.009	0.072	*F*_(6,232)_ = 23.599	<0.001
	Sexual Violence	0.171	2.683	0.008					
	Relational Violence	0.173	2.636	0.009					
	Physical Violence	−0.062	−0.946	0.345					
	Verbal Violence	0.075	1.111	0.268					
	Conflict Resolution	0.098	1.81	0.072					

Model 1—Jealousy subscale and subscales of the Ambivalent Sexism Inventory (ASI); Model 2—Jealousy subscale and subscales of the Conflict in Adolescent Dating Relationships Inventory (CADRI).

## Data Availability

The datasets generated during and/or analysed during the current study are available from the corresponding author on reasonable request.

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
