# Peer review of "Jealousy, Violence, and Sexual Ambivalence in Adolescent Students According to Emotional Dependency in the Couple Relationship"

_children, 2021, doi:10.3390/children8110993_

Round 1

Reviewer 1 Report

This would be a stronger study if there was a theoretical background that linked together the variables that were studied.

A general issue is that there are places when more causal language is used than warranted:

-In second paragraph of the introduction first sentence the verb “generates” might be too causal to describe the relationship between variables

-Causal language is also present in the third paragraph of the introduction

-In the first sentence on p. 10 the word “effect”

Some more specific issues:

The take-away message about the results could be more clear in the abstract

The link between ambivalent sexism and the other variables is not explained very clearly in the introduction (and are the ideas presented gender-specific?)

This is a fairly small sample of boys – the statistical power should be determined

Were data used only from those with a partner? (If so, a very small sample size and statistical power is not likely sufficient)

Can the PEDS be filled out regardless of present partner status?

How was the study described to participants?

The results that are presented in tables do not need to be repeated in the text of the results section.

In the discussion should the participants be described as girls and boys rather than men and women?

Author Response

REVIEWER 1

We would like to thank the reviewer for his comments and suggestions, which have been accepted and will undoubtedly improve the final result. We hope we have met the reviewer's expectations.

Q: A general issue is that there are places when more causal language is used than warranted:

A: The reviewer's suggestion has been accepted and, in general, the text has been adjusted. Efforts have been made to suggest and not to maintain such a causal line of argumentation. This "causal language" has not been modified and has been maintained in the citations of previous studies whose authors have considered it to be so.

Q: In second paragraph of the introduction first sentence the verb “generates” might be too causal to describe the relationship between variables

A: The reviewer's suggestion has been accepted and, in general, the text has been adjusted. Efforts have been made to suggest and not to maintain such a causal line of argumentation. This "causal language" has not been modified and has been maintained in the citations of previous studies whose authors have considered it to be so.

Q: Causal language is also present in the third paragraph of the introduction

A: The reviewer's suggestion has been accepted and, in general, the text has been adjusted. Efforts have been made to suggest and not to maintain such a causal line of argumentation. This "causal language" has not been modified and has been maintained in the citations of previous studies whose authors have considered it to be so.

Q: In the first sentence on p. 10 the word “effect”

A: The reviewer's suggestion has been accepted and the text has been adjusted.

Q: The take-away message about the results could be more clear in the abstract.

A: The wording has been improved, a brief conclusion has been included and now we hope that the abstract will be clearer in the presentation of the results.

Q: The link between ambivalent sexism and the other variables is not explained very clearly in the introduction (and are the ideas presented gender-specific?).

A: An extension of the text has been made, in the introduction, where sexual ambivalence, violence, couple relationship, adolescents, etc. are mentioned. We believe that the reviewer's observation has been taken into account.

Q: This is a fairly small sample of boys – the statistical power should be determined.

A: The calculation to determine the sample size has been indicated in the section on participants.

Q: Were data used only from those with a partner? (If so, a very small sample size and statistical power is not likely sufficient).

A: No, students who were not currently in a dating relationship also participated (inclusion criteria: currently in a dating relationship of at least one month's duration; or recently in a dating relationship, with no more than two months since the breakup). The tests can be completed, according to limitations (see next comment) as long as the relationships have been recent.

Q: Can the PEDS be filled out regardless of present partner status?.

A: This test (Partner's Emotional Dependency Scale-PEDS) can be completed even if there is no current partner, as long as there has been a recent relationship. Therefore, one of the inclusion criteria was: having a current relationship of at least one month's duration; or having been in one recently, with no more than two months since the breakup.

Q: How was the study described to participants?.

A: In the procedure, the information given to participants has been included.

Q: The results that are presented in tables do not need to be repeated in the text of the results section.

A: If we have not made a mistake when reviewing the work, it is only in table 6 where some data is repeated in the text. In this specific case (7 data have been repeated), the aim was to highlight relevant information in the table that sought to cover one of the objectives of the work. Our approach would be to keep it this way, but if the reviewer considers it feasible, it can be eliminated from the text.

Q: In the discussion should the participants be described as girls and boys rather than men and women?

A: The comment is appreciated and has been considered in the text of the abstract, the discussion and at the end of the introduction when the hypotheses are stated. Men/women have been replaced by boys/girls in the cases that referred to our study. In the works of other authors that have been cited, the proposal of the referenced author has been maintained.

Reviewer 2 Report

  • Authors are advised to comment in the abstract that different questionnaires validated have been filled in or completed by adolescents. As the way the results have been presented has created the doubt where they came from.
  •  
  • The manuscript states the aim of the study, but does not discuss any hypotheses. However, several hypotheses are answered in the discussion. In order to be able to answer the hypotheses, they must be described in advance, together with the objective of the study.
  • The age of the adolescents or the age range is unknown. It is said that there are a number of minors, but this is not specified. These issues should be specified.

  • How the sample size was obtained is not discussed, and this is a very important issue. Nor is it explained from which classes the sample has been selected, nor how the sampling has been done.

  • The inclusion and exclusion criteria should be explained.

  • Has this study passed the evaluation of any Ethics Committee? This information should be added in the manuscript.

  • The answers to the hypotheses are not detailed in the discussion, all comments made should be withdrawn.

  • The conclusions section should be created to respond to the objective of the study and whether it is considered appropriate to the hypotheses formulated.

  • The bibliography must be updated very carefully; there are very old references that must be reconsidered and specified, as they are much more recent.

Author Response

We would like to thank the reviewer for his comments and suggestions, which have been accepted and will undoubtedly improve the result. We hope we have met the reviewer's expectations.

Q: Authors are advised to comment in the abstract that different questionnaires validated have been filled in or completed by adolescents. As the way the results have been presented has created the doubt where they came from.

A: The reviewer's suggestion has been accepted and the information on the instruments used has been included in the Abstract.

Q: The manuscript states the aim of the study, but does not discuss any hypotheses. However, several hypotheses are answered in the discussion. In order to be able to answer the hypotheses, they must be described in advance, together with the objective of the study.

A: At the end of the Introduction, after the objective of the study, the four working hypotheses that are subsequently analyzed in the discussion are specified.

Q: The age of the adolescents or the age range is unknown. It is said that there are a number of minors, but this is not specified. These issues should be specified.

A: In the Participants section and the first paragraph (first line) of the Results, the mean age was already indicated and the Minimum and Maximum age has been included. The percentage of participants according to the recognized age has also been included.

Q: How the sample size was obtained is not discussed, and this is a very important issue. Nor is it explained from which classes the sample has been selected, nor how the sampling has been done.

A: The procedure included information on how the various classes were visited and the criteria for doing so. It has already been indicated that the sampling was incidental non-probabilistic (The schools were selected by non-probabilistic sampling according to accessibility and availability criteria). In the section on participants, the calculation to determine the sample size has been indicated.

Q: The inclusion and exclusion criteria should be explained.

A: Inclusion criteria for the study have been included in the procedure.

Q: Has this study passed the evaluation of any Ethics Committee? This information should be added in the manuscript.

A: In the section "Declaration of interest statement" before the References it was specified that it was submitted to the Bioethics Committee of the University. However, it has been included in the procedure and the date on which it was submitted to the Committee (05-July-2020).

Q: The answers to the hypotheses are not detailed in the discussion, all comments made should be withdrawn.

A: The reviewer's comment is gratefully acknowledged. We understand that the objectives and hypotheses are usually answered and analyzed in the Discussion section; for this purpose, articles published in various journals, included Children journal, in recent years, have been reviewed and found to be so. Therefore, our preference is that you maintain the position in the discussion.

Q: The conclusions section should be created to respond to the objective of the study and whether it is considered appropriate to the hypotheses formulated.

A: Bearing in mind the reviewer's comments, and as indicated in the previous comment, a brief section on conclusions has been created.

Q: The bibliography must be updated very carefully; there are very old references that must be reconsidered and specified, as they are much more recent.

A: The text has been expanded, with new and updated citations. Those of relevant works on the subject have been maintained, even though they are very old.

Round 2

Reviewer 1 Report

The authors have made the changes I suggested resulting in an improved manuscript.

Author Response

Thank you very much. Thanks to your suggestions we believe that the work has improved. 

In relation to the recommendations of another reviewer, some references have been updated.  You will find them marked in red. 

Best regards,

Reviewer 2 Report

Changes have been made in the manuscript that have been commented to the authors, but they are asked to review the bibliography again. There are very old bibliographic references that can be updated.

Author Response

Again, we would like to thank the reviewer for his comments and suggestions, which have been accepted and will undoubtedly improve the final result. We hope we have met the reviewer's expectations.

In relation to your recommendations, some references have been updated.  You will find them marked in red. 

We hope that this update is the right one. 

Best regards,
